# Research on Internal Shape Anomaly Inspection Technology for Pipeline Girth Welds Based on Alternating Excitation Detection

**DOI:** 10.3390/s23177519

**Published:** 2023-08-30

**Authors:** Rui Li, Pengchao Chen, Jie Huang, Kuan Fu

**Affiliations:** 1Pipechina Research Institute, Langfang 065000, China; chenpc@pipechina.com.cn (P.C.); fukuan@pipechina.com.cn (K.F.); 2College of Mechanical and Transportation, China University of Petroleum, Beijing 102249, China; 2022315301@student.cup.edu.cn

**Keywords:** oil and gas pipelines, internal inspection, girth weld formation abnormalities, system development, alternating excitation detection

## Abstract

Abnormal formation of girth weld is a major threat to the safe operation of pipelines, which may lead to serious accidents. Therefore, regular inspection and maintenance of girth weld are essential for accident prevention and energy security. This paper presents a novel method for inspecting abnormal girth weld formation in oil and gas pipelines using alternating excitation detection technology. The method is based on the analysis of the microscopic magnetic variations in the welded area under alternating magnetic fields. An internal inspection probe and electronic system for detecting abnormal girth weld formation were designed and developed. The system’s capability to identify misalignment, undercutting, root concavity, and abnormal formation height of girth weld was tested by numerical simulation and experimental study. The results show that the detection system can effectively identify a minimum misalignment of 0.5 mm at a lift-off height of 15 mm. The proposed method offers several advantages, such as rapid response, low cost, non-contact operation, and high sensitivity to surface flaws in ferromagnetic pipelines.

## 1. Introduction

In recent years, global energy policies and discussions on energy topics have been centered around addressing the “three elements of energy challenge”: safety, affordability, and lower carbon emissions [1,2]. Pipelines, characterized by their large transport capacity, long distances, and low energy consumption, are not only responsible for domestic transportation of oil and natural gas products but also gradually realizing the long-distance and large-scale transport of hydrogen and carbon dioxide, thereby supporting global energy transition and decarbonization. Therefore, the position of pipeline transportation as a national energy infrastructure will remain unchanged and will continue to receive strong policy support [3].

As of the end of 2022, the total mileage of active oil and gas pipelines worldwide is approximately two million kilometers, with 26,708 km of pipelines under construction, and the total millage is expected to reach 2,193,350 km by 2025. According to data from the Pipeline and Hazardous Materials Safety Administration (PHMSA) of the U.S. Department of Transportation, there have been 12,785 significant pipeline accidents in various states of the United States from 2003 to 2022 [4]. These accidents resulted in 274 deaths and 1120 injuries. Over these 20 years, the average annual cost of these accidents was $541 million, with a total loss of approximately $10.8 billion. This indicates that the demand for pipeline inspection will further expand to prevent environmental pollution and human casualties caused by serious accidents such as pipeline leaks and explosions [5].

According to the latest report from the European Gas Pipeline Incident Data Group (EGIG), pipeline discontinuities are mainly caused by the following three reasons: corrosion (38%), construction defects/metal failures (19%), and external interference (17%). With the significant improvement in manufacturing level, defects generated during the pipeline prefabrication process are rarely discovered, and construction defects/metal failures mainly occur at girth weld, with corrosion accounting for the largest proportion. Some accidents are also related to girth weld defects [6,7,8].

These defects significantly reduce the strength of the pipelines. Common girth weld defects include cracks, misalignment, undercut, and incomplete penetration. As the most effective method for maintaining pipeline integrity and safety, internal inspection of pipelines is an important means to determine their operating condition and ensure their operational capability [9]. Existing pipeline internal inspection mainly relies on nondestructive testing techniques, aiming to collect pipeline characteristic information without damaging the internal structure of the pipelines [10]. By utilizing existing technologies and equipment and conducting subsequent data processing, these techniques characterize the defects in the pipeline body. Eddy current testing, with its high detection rate for volumetric defects, low susceptibility to external interference, and fast detection speed, can effectively address pipeline failures caused by corrosion [11], especially suitable for rapid inspection of large-area and long-distance pipelines, thereby compensating for the limitations of conventional inspection methods in detecting welding anomalies [12].

As the most common electromagnetic inspection method for conducting elements, Eddy current testing has been widely applied in detecting surface defects, subsurface corrosion, and coating characterization of conductive materials [13,14,15]. In 1879, British engineer David Hughes utilized the eddy current effect to achieve the sorting of metallic materials, marking the first practical application of eddy current theory in engineering [16]. In 1933, German scientist Foerster, aiming to reduce interference during the operation of inspection equipment at that time, employed complex plane signal analysis for signal analysis of the detection instrument, establishing the two-dimensional impedance plane analysis method for eddy current testing signals [17]. This greatly improved the signal-to-noise ratio of detection signals and further promoted the engineering applications of eddy current testing, laying the foundation for modern conventional eddy current testing. With the continuous development of eddy current testing theory, materials science, and digital signal processing technology, various techniques have been developed, including multi-frequency eddy current testing, remote field eddy current testing (RFEC), eddy current array (ECA) testing, and pulsed eddy current testing (PEC) [13]. Currently, mature eddy current internal inspection equipment abroad includes the RoCorr IEC eddy current internal detector by ROSEN, the RoMat DMG magnetic leakage and eddy current composite internal detector, and the See Snake remote field eddy current internal detector developed by Russell [18].

In China, since the 1980s, domestic universities and research institutes have conducted extensive research on flaw detection in oil and gas pipelines using eddy current testing technology, primarily focusing on discrimination of defects on the inner and outer walls of channels and extraction of crack and deformation defect information. Research in impedance analysis of inline coils, optimization of induction coil parameters, and simulation techniques has also made significant progress. Among them, the research team led by Professor Wu Xinjun from Huazhong University of Science and Technology [16,19,20] applied pulsed eddy current testing technology to detect corrosion defects in pipelines by measuring the attenuation of eddy currents within the pipeline. The research team led by Professor Luo Feilu from the National University of Defense Technology [21,22] utilized multi-frequency eddy current testing technology to process the spectral energy of eddy current sensor output signals for quantitative analysis of defect length. Xu Zhiyuan and colleagues from Xiangtan University employed magnetic field focusing and rotating magnetic field advantages to propose a rotating focused field eddy current sensing technology for detecting defects of arbitrary orientations, enabling defect detection in any direction and quantitative evaluation of defect depth using signal amplitude [23].

The rest of this paper is organized as follows. Section 2 introduces the deviation and movement of magnetic domains in the welded area under alternating magnetic fields. Section 3 numerically simulates abnormal models of girth weld formation in pipelines, investigating the variation of height abnormal response signals in girth weld formation. Section 4 describes the composition and design principles of the electronic system for detecting girth weld formation defects in pipelines. Section 5 presents the experimental research results of specimens with girth weld formation abnormalities and compares them with simulation results. Finally, Section 6 summarizes the existing conclusions and achievements.

## 2. Theoretical Analysis of Girth Weld Forming Inspection

The alternating excitation detection technique is developed based on the principles of eddy current testing. Its fundamental principle is to identify surface defects of specimens by analyzing the microscopic magnetic variations in the tested material [24]. In this chapter, starting from the mechanism of magnetic field generation in ferromagnetic materials and combining it with the basic principles of eddy current testing, we focus on analyzing new methods and theoretical foundations of alternating excitation detection for surface defects in ferromagnetic pipelines. This analysis also provides support for subsequent numerical simulations and experimental investigations.

### 2.1. Magnetization Mechanism of Ferromagnetic Materials

A magnetic domain refers to a small magnetized region within a ferromagnetic material where atomic magnetic moments spontaneously align due to the interaction of electron spins. As illustrated in Figure 1, in the absence of magnetization, adjacent magnetic domains within ferromagnetic materials exhibit different orientations of atomic magnetic moments that mutually cancel out, resulting in a lack of observable magnetism [25].

When a ferromagnetic material is subjected to an external magnetic field, as shown in Figure 2, the magnetic moments within its internal magnetic domains gradually align with the external field, and the domain walls continuously move. This results in an increasing number of domains aligned with the external field and a decreasing number of domains opposed to it. Eventually, all magnetic domains align their magnetic moments with the external field, reaching a state of saturation for the material itself [26,27].

### 2.2. Alternating Current Magnetization Inspection Technology

Alternating current magnetization inspection (ACMI) is a method that has evolved from eddy current testing. This technique is based on the principles of Faraday’s electromagnetic induction, Ampère’s circuital law, and the theory of magnetic domain deviation and domain wall displacement in ferromagnetic materials. ACMI offers several advantages, including rapid response, low cost, and non-contact operation. It demonstrates excellent defect detection capability for surface flaws in ferromagnetic materials such as oil and gas pipelines. As a result, ACMI is a promising nondestructive testing technology with wide-ranging applications [28].

ACMI principle is depicted in Figure 3. When an alternating current, *I*_1_, flows through the coil, it generates an alternating magnetic field, *H*_1_, perpendicular to the surface of the specimen based on Ampere’s law. According to Faraday’s law of electromagnetic induction, this magnetic field induces eddy currents (EC) on the surface of the specimen. Simultaneously, the eddy current magnetic field, *H*_3_, is excited to impede the variations in the alternating magnetic field, *H*_1_. The magnitude of *H*_2_ can be calculated using Equations (1) and (2).
(1)$$\nabla \times \left(\frac{A}{\mu }\right)=J_{0}+J_{e}$$

In Equation (1), *μ* denotes the magnetic permeability of the tested metal, *A* signifies the magnetic vector potential, *J*_0_ corresponds to the current density within the excitation coil, and *J_e_* stands for the induced eddy currents within the tested metal specimen, which can be mathematically expressed as:(2)$$J_{e}=-\sigma \frac{\partial A}{\partial t}+\nabla V$$
where *V* represents voltage in Equation (2). Under the influence of the alternating magnetic field *H*_1_, the internal magnetic domains within the metal undergo rotation, inducing a magnetization magnetic field *H*_2_ that aligns with the direction of the alternating magnetic field *H*_1_. The magnetic sensor positioned at detection point *A*_1_ detects a composite magnetic flux signal *B* in the surrounding space, which can be determined by Equation (3).
(3)$$\vec{B}=\vec{B_{1}}+\vec{B_{2}}+\vec{B_{3}}$$

Within Equation (3), *B* represents the magnetic flux density of the composite magnetic field. *B*_1_ corresponds to the magnetic flux density of the excitation’s alternating magnetic field, *H*_1_, and can be computed using the Biot-Savart law under known excitation parameters. On the other hand, *B*_2_ stands for the magnetic flux density of the magnetization magnetic field, *H*_2_, and can be determined using Equation (4) [29]. Lastly, *B_3_* denotes the magnetic flux density of the eddy current magnetic field, *H_3_*, which can be evaluated using Equation (5).
(4)$$B_{2}=\frac{\mu_{0}J_{C}}{2}\int_{0}^{\infty }\frac{\chi \left(\lambda_{0}r_{1},\lambda_{0}r_{2}\right)}{\lambda_{0}^{2}}(e^{-\lambda_{0}d}-e^{-\lambda_{0}(d+h)})R(\lambda_{0})e^{-\lambda_{0}y}\left[J_{1}(\lambda_{0}x)x_{0}+J_{0}(\lambda_{0}x)y_{0}\right]d\lambda_{0}$$

In Equation (4), *μ*_0_ represents the magnetic permeability of a vacuum. *J*_c_ denotes the equivalent current density of an ideal single-turn coil. *J*_1_(*λ*_0_) represents the first-order Bessel function. *λ_0_* is the integration variable, and *R*(*λ_0_*) is a physical quantity associated with the electrical conductivity and magnetic permeability of the material under test. *H*_1_ corresponds to the magnetic field intensity generated by the excitation coil at the specimen location. *d* represents the distance between the ring and the specimen. *r*_1_ represents the inner diameter of the excitation coil; r_2_ denotes the outer diameter of the excitation coil.

The magnetization induction intensity, denoted as *B_3_*, can be expressed as follows:(5)$$B_{3}=\{\begin{array}{cc} \mu_{0}\left(H_{1}+\chi H_{g}\right) & H_{g}=H_{w}\\ \mu_{0}\left[H_{1}+\chi \left(H_{w}-H_{y}\right)\right] & otherwise \end{array}$$

In Equation (5), *χ* represents the magnetization susceptibility. *H*_1_ denotes the magnetic field intensity generated by the excitation coil. *H_g_* represents the magnetic field intensity of the tested component. *H_w_* represents the magnetic field intensity in the presence of a defect in the tested piece. *H_y_* represents the decrease in magnetic field intensity caused by the defects in the tested component.

When there is a weld flaw in the local pipeline under inspection, the alternating magnetic field *H*_1_ remains unchanged. However, the presence of the defect impedes the formation of an eddy current loop, leading to an increase in the magnetic flux density corresponding to the localized eddy current field, resulting in an increased magnetization magnetic field *H*_2_. Meanwhile, the occurrence of weld defects, misalignment, biting edges, or surface concavities causes a decrease in the eddy current magnetic field *H*_3_. Overall, the presence of weld defects causes distortion (reduction) of the composite magnetic flux in space. The reverse solution of key dimensions of detected pipeline girth weld defects can be achieved by extracting characteristic quantities from the transformed distortion signal.

## 3. Finite Element Simulation Analysis of Girth Weld Forming Anomalies

Currently, various techniques are employed to solve electromagnetic field problems, including analytical methods, numerical simulation methods, and experimental methods. Analytical methods often rely on simplified analytical models to tackle complex issues, leading to significant errors and intricate computational processes. On the other hand, experimental methods offer calculations that closely resemble real-world conditions. However, they are subject to constraints imposed by experimental costs and conditions. Consequently, prior to conducting experimental validation, it is necessary to perform numerical simulations to replicate the actual working conditions accurately [30].

### 3.1. Finite Element Simulation Model

The finite element method (FEM) is a highly efficient numerical computation technique widely employed for solving complex systems with intricate physical properties and geometric shapes. During engineering practice, it is commonly used to analyze and simulate continuous physical systems. In this study, the widely used low-frequency electromagnetic field finite element software Maxwell 2018 R2, commonly used in engineering electromagnetics, is utilized for numerical analysis of typical girth weld defects [31].

To begin with, a simplified modeling approach was employed for the detection of anomalies in girth weld formations using the eddy current field module within Maxwell’s magnetic field analysis. The geometric parameters are delineated in Table 1. Given that the curvature of the 1219 mm pipe surpasses that of the excitation coil by a factor of five, the curvature’s impact on the detection signal is negligible. Hence, the pipe’s intricate geometry was approximated using a flat plate model [18]. Given the substantial computational time required for three-dimensional solutions and the circumferential symmetrical arrangement of probes along the pipe, the establishment of a two-dimensional (2D) model is a prevalent electromagnetic simulation simplification technique, as depicted in Figure 4. Moreover, to enhance the passing capability of the internal detector within the variably thick 1219 mm pipe, a lift-off height (TL) of 15 mm was adopted between the probe and the inspected pipe.

Material properties are assigned to different parts of the simulation model. The electrical conductivity of air is set to 0 S/m. However, in Maxwell’s differential calculations, the conductivity appears in the denominator, which can lead to non-convergence. To avoid this issue, the electrical conductivity of air is set to 1 S/m. The material parameters of the simulation model are presented in Table 2.

Influenced by the skin effect, current tends to concentrate on the surface of the test specimen, necessitating the refinement of the mesh within the skin depth and the surrounding air domain. The eddy current module within Maxwell’s magnetic field analysis is equipped with adaptive meshing capabilities, which can mitigate the impact of mesh configuration on solution outcomes. However, due to the limitation of mesh shapes to tetrahedral elements and the inability to perform operations like mapping and sweeping, manual mesh partitioning is still required for regions where mesh refinement is necessary.

In this study, excitation was conducted using a sinusoidal AC current with an amplitude of 1A and a frequency of 1 kHz, with natural boundary conditions set. Specific solution parameters are detailed in Table 3.

### 3.2. Analysis of Abnormal Signals in the Forming Height

To investigate the influence of girth weld height on the magnetic field signal, the girth weld height (semi-circular radius) is sequentially set to 0.5 mm, 1 mm, 1.5 mm, and 2 mm. The output results of vertical magnetic field strength, *B_y_*, and Horizontal magnetic field strength, *B_x_*, at the central position *A*_1_ (Figure 4) of the probe are obtained and shown in Figure 5 and Figure 6.

Figure 5 reveals that during the axial scanning of the probe along the pipeline, a signal peak in *B_y_* occurs when passing through the girth weld region, and as the height of the weld formation increases, the extent of signal protrusion also progressively amplifies. The peak point of the signal corresponds to the centre position of the girth weld protrusion. As the girth weld height increases from 0.5 mm to 2 mm, the peak variation of *B_y_* reaches 1.41 Oe, indicating a significant change in *B_y_* with increasing girth weld height.

Figure 6 demonstrates that during the axial scanning of the probe along the pipeline, when not passing through the girth weld region, the *B_x_* signal component remains at 0. As the probe passes through the girth weld region, the *B_x_* signal exhibits a “decrease-increase-decrease” trend. Additionally, the peak and valley values of the *B_x_* signal correspond to the center position of the girth weld protrusion, showing a noticeable variation with changes in girth weld height.

By examining the magnetic field intensity signals in both horizontal and vertical directions, it is evident that with the continuous increase in weld height, the differences between the signals progressively magnify. There exists a strong correlation between the height of the circumferential weld and the magnetic field signals. This provides a foundational basis for the subsequent experiments, affirming the feasibility of investigating pipe defects through the analysis of pipe eddy current distribution and spatial magnetic field disturbances.

## 4. Development of Sensing and Electronic Systems

### 4.1. Probe Design

The pipeline girth weld formation defect detection system mainly consists of a detection probe, an interface box, a data acquisition and storage unit, and a power system. The detection probe, arranged in a circumferential array along the pipeline, is primarily responsible for converting the abnormal girth weld formation into corresponding analogue signals.

The girth weld formation detection probe, as shown in Figure 7, is constructed with enamelled wire wound into an excitation coil. Under the excitation of the current signal generation module, it generates an alternating magnetic field. The TMR (Tunneling Magnetoresistance) magnetic sensor is a novel type of magnetic sensor that consists of two layers of ferromagnetic material (free layer/fixed layer) sandwiching a thin insulating barrier layer of 1–2 nm. When the magnetization direction of the fixed layer is fixed, the magnetization direction of the free layer changes according to the external magnetic field, leading to a change in the resistance of the device. The TMR magnetic sensor captures the magnetic field signal and converts it into a voltage signal output.

The model of the magnetic sensor is TMR2503, which produced by Jiangsu Duowei Technology in China. Under a 5 V power supply, the magnetic field detection sensitivity reaches 5 × 10^−4^ V/Oe. It senses the magnetic field information perpendicular to the TMR element and has a linear sensing range of ±750 Oe with a background noise of 5 Oe. The coil frame and packaging shell are both made of resin as the raw material and manufactured using 3D printing technology with photopolymerization. The packaging shell is the outermost layer of the detection probe and provides protection and support for the excitation coil and the magnetic sensor. The coil frame holds the excitation coil and the magnetic sensor.

### 4.2. Design of Signal Generation Module

The internal components of the interface box encompass the current signal generation module and the signal analog-to-digital conversion module. Each individual current signal generation module is responsible for driving ten detection probes to induce alternating magnetic field signals in space. The welding deformation anomaly detection probes require a sinusoidal power-type signal input with a frequency of 1 kHz. To ensure the smoothness and integrity of the sinusoidal waveform, each sine wave consists of a hundred data points, with a frequency of signal point updates reaching 100,000 Hz. The circuit module for generating sinusoidal signals is depicted in Figure 8, employing the STM32H743VIT6 as the chosen core microcontroller. This high-performance ARM Cortex-M7 MCU is equipped with DSP and DP-FPU functionalities. It boasts 2 MB Flash memory, 1 MB RAM, a 480 MHz CPU, ART accelerator, first-level cache, external memory interfaces, and a plethora of peripheral capabilities. The Cortex-M7 core includes a floating-point unit (FPU) for precision, supporting Arm double-precision (in compliance with IEEE 754 standards) and single-precision data processing instructions and data types. The STM32H7 MCU supports a complete set of DSP instructions and a memory protection unit (MPU), enhancing the security of applications.

Experimental measurements have revealed that the signal point update frequency reaches 200 kHz, effectively fulfilling the operational requisites of the signal generation module. This frequency ensures the adequacy of the sine wave generation process. Furthermore, adjustments to the amplitude and frequency of the sinusoidal waveform can be made through communication or alternative means. Concurrently, in accordance with the specified sampling phase, the output sampling trigger command is initiated. The microcontroller employs its inherent DAC functionality and is complemented by a high-frequency timer to generate DAC signals that conform to the sinusoidal waveform. Through the utilization of biasing and amplification circuits, the sinusoidal voltage signal produced by the microcontroller is converted into a voltage signal based on 0 V. Subsequent conditioning of this signal is performed by operational amplifiers, while concurrently undergoing amplification via current acquisition to establish a closed-loop output governed by current control. Ultimately, this culminates in the realization of a current-driven waveform generator.

### 4.3. Design of Signal Transmission Module

The analog-to-digital conversion module is responsible for transforming the analog voltage signals transmitted by the girth weld formation detection probes into digital signals for parallel signal transmission. The detection signals directly outputted by the TMR magnetic sensitive element are exceedingly weak and often contaminated with noise signals, which hampers effective detection. Therefore, an amplification and filtering circuit is devised to amplify and filter the detection signals, thereby enhancing defect signal discernibility. The schematic of the amplification and filtering circuit is illustrated in Figure 9.

This amplification and filtering circuit employs the low-noise operational amplifier SGM722 and high-precision capacitors and resistors. For the experimentation, a sinusoidal AC excitation at 500 Hz is employed. The high-pass cutoff frequency of the amplification and filtering circuit is set at 200 Hz, while the low-pass cutoff frequency is set at 1000 Hz, resulting in an amplification factor of 324. The output voltage after amplification and filtering can be mathematically represented as follows:(6)$$V_{OUT}=V_{{}_{REF}}+Av\cdot \left(V_{{}_{0+}}-V_{{}_{0-}}\right)$$

In the provided equation, the variables are defined as follows: $V_{OUT}$ represents the output voltage (*V*); $V_{{}_{REF}}$ represents the reference voltage (*V*); Av represents the amplification factor; $V_{{}_{0+}}$ represents the analog positive output voltage (*V*) of the magnetic sensitive element’s differential mode; $V_{{}_{0-}}$ represents the analog negative output voltage (*V*) of the magnetic sensitive element’s differential mode. By adjusting the RC values in the circuit, it is possible to achieve the desired changes in the conditioning circuit’s bandwidth and center frequency. Simultaneously, this adjustment can also modify the circuit’s amplification factor, ensuring that the analog signal output falls within a collectable range and exhibits a distinct analog response to girth weld formation anomalies. The data acquisition module is responsible for the parallel collection and storage of multi-channel digital signals. The power supply module ensures the system’s overall power provision. The cables are tasked with transmitting signals and energy between the sensors and the electronic system.

The data acquisition module performs similar acquisition and storage of digital signals from multiple channels. The power module is responsible for supplying power to the entire system. The cables transmit signals and energy between the sensing and electronic systems.

## 5. Experimental Study on Detection of Girth Weld Forming Anomalies

### 5.1. Experimental Platform Setup

As shown in Figure 10, a three-axis coordinated experimental platform was constructed to enable the linkage of the *x*, *y*, and *z* axes. The effective travel distances in the *x*, *y*, and *z* directions are 1500 mm, 900 mm, and 300 mm, respectively. The stepper motor can reach a maximum speed of 930 mm/s under no-load conditions and 500 mm/s under full load conditions, with a module precision of 0.02 mm.

This experiment utilized artificially fabricated welding defect specimens, encompassing four typical welding defects: misalignment, abnormal formation height, root concavity, and undercut. Among these, the misalignment defect was implemented on a flat specimen made of 45# steel. The abnormalities in formation height, root concavity, and undercut were conducted on 1/4 pipe specimens composed of X52 material.

### 5.2. Analysis of Girth Weld Formation Abnormal Signals

#### 5.2.1. Misalignment Detection Experiment

To investigate the detection capability of the developed girth weld formation probe for misalignment and the variation of response signals with the amount of misalignment, the experimental setup involved scanning specimens with misalignment values of 0.5 mm and 1 mm at distances TL between the probe and the tested material set at 5 mm, 10 mm, and 15 mm, as shown in Table 4.

From Figure 11 and Figure 12, it can be observed that as the probe scans from the left side to the right side of the welded pipe, the output quantity of the probe increases, and the signal experiences distortion, with the degree of distortion closely related to the lift-off height and the amount of misalignment. When the TL value of the probe increases from 5 mm to 15 mm, the distortion values of the probe’s output signal for misalignment decrease by 93.87% and 92.51%, respectively. Thus, it can be seen that for scans with the same amount of misalignment, the distortion of the probe’s output signal decreases with increasing TL height. Similarly, at the same TL height, the degree of distortion in the probe’s output signal increases with an increase in misalignment.

#### 5.2.2. Undercut Detection Experiment

Based on Figure 13, it can be observed that when the probe passes through the welded undercut region, the signal at the root exhibits a concave shape, and the depth of this concavity is related to the size of the undercut. As shown in Table 5 generally, a larger undercut area leads to a more pronounced signal distortion, resulting in a deeper dip in the signal at the weld root. However, due to the weld region’s complex wall morphology and the undercut’s small extent, the distortion of the signal is significantly influenced by the lift-off height and vibrations. As a result, quantitative inversion of the undercut size in the weld detection process poses certain challenges.

#### 5.2.3. Weld Forming Height Anomaly Detection Experiment

Figure 14 shows that during the axial scanning along the pipeline, a protrusion in the signal is generated when passing through the region with welding-induced anomalies, and the degree of signal protrusion varies with the height of the welding anomaly. This observation aligns with the simulation outcomes. The distortion values ΔV of the response signals for welding patterns numbered ①, ②, ③, and ④ are 0.1751 V, 0.2230 V, 0.0442 V, and 0.2653 V, respectively. Due to the influence of machining errors on the test specimen, the baseline values on the inner surface of the pipeline differ, resulting in the welding-induced signals not exhibiting the expected regular changes. However, it can be observed that different heights of welding anomalies lead to varying responses in the height anomaly signal. When the circumferentially arranged welding detection probes operate inside the pipeline, it is expected that the challenging task of detecting height anomalies in the welding patterns can be addressed through comparative analysis of the signals from the circumferentially arranged probes.

#### 5.2.4. Girth Weld Root Concavity Detection Experiment

From Figure 15, it can be seen that during the axial scanning of the probe along the pipeline, the signal fluctuates when passing through the concave region at the root of the girth weld; however, no apparent systematic changes are observed in the probe response.

One reason for this can be attributed to the fact that the welding-induced anomaly detection probe is based on the principle of domain magnetization and detects the vector and extent of magnetization deviation within a certain range on the inner wall of the pipeline. Moreover, the complex electromagnetic characteristics of the girth weld region are influenced by the presence of “fish-scale” patterns on the surface of the weld and the heat-affected zone. Therefore, the ability of this probe to identify concave defects at the root of the girth weld requires further improvement.

Experimental investigation of weld formation anomaly detection technology for oil and gas pipelines leads to the following conclusions:(1)The distortion of the probe output signal decreases as the lift-off (TL) height increases for scans with the same misalignment. Conversely, for scans with the same TL height, the degree of distortion of the probe’s output signal increases with misalignment.(2)As the undercut region grows, the signal distortion becomes more pronounced, resulting in a deeper signal drop at the girth weld’s root. However, due to the girth weld area’s complex wall morphology and the undercut’s limited extent, the signal distortion is significantly influenced by the lift-off height and vibration. Consequently, achieving quantitative inversion of the undercut during the detection process presents certain challenges.(3)The developed detection probe effectively detects misalignment of 0.5 mm at TL values of 15 mm. Additionally, it exhibits specific capability in identifying anomalies in girth weld forming height and undercuts. However, further enhancements are required to detect concave defects at the girth weld’s root.

## 6. Conclusions

This article presents a novel method for detecting abnormal girth weld formation in oil and gas pipelines using alternating excitation detection technology. The method is based on the analysis of the microscopic magnetic variations in the welded area under alternating magnetic fields. An internal detection sensor and electronic system for detecting abnormal girth weld formation were designed and developed. The detection system’s capability to identify misalignment, undercutting, root concavity, and abnormal formation height of girth welds was tested by numerical simulation and experimental study. The results show that the detection system can effectively identify a minimum misalignment of 0.5 mm at a lift-off height of 15 mm. The proposed method offers several advantages, such as rapid response, low cost, non-contact operation, and high sensitivity to surface flaws in ferromagnetic pipelines. The article also discusses the theoretical analysis, finite element simulation, and system development of the proposed method. The article aims to contribute to the field of pipeline inspection and maintenance by providing a new technique for detecting girth weld defects.

## Figures and Tables

**Figure 1 sensors-23-07519-f001:**
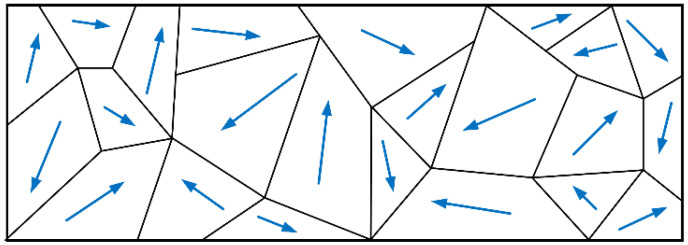
Magnetic domain.

**Figure 2 sensors-23-07519-f002:**
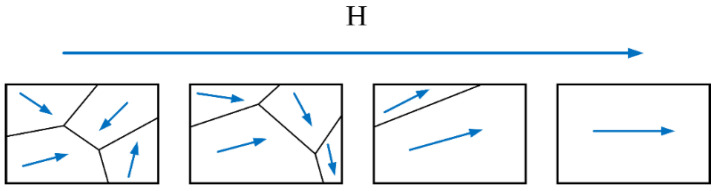
Domain magnetization process.

**Figure 3 sensors-23-07519-f003:**
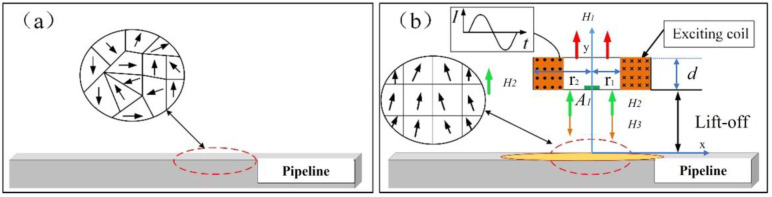
Schematic diagram of space magnetic field gradient detection technology. (**a**) Magnetic domains and domain wall state without alternating magnetic field. (**b**) Magnetic domain deflection and domain wall displacement under alternating magnetic Field.

**Figure 4 sensors-23-07519-f004:**
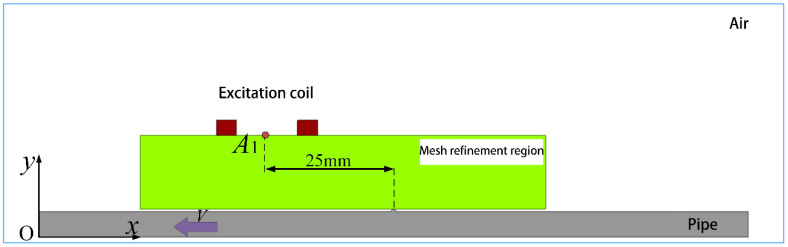
Simplified simulation model.

**Figure 5 sensors-23-07519-f005:**
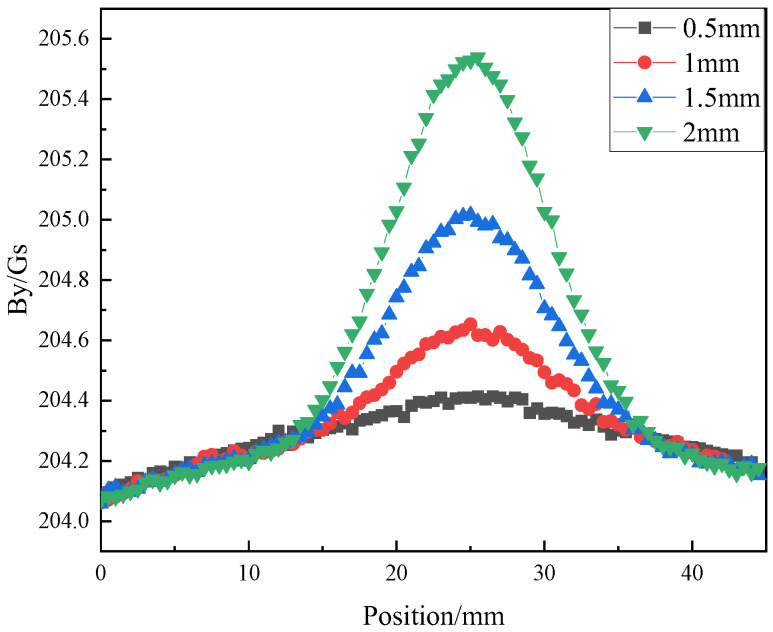
The relationship between girth weld forming height and *B_y_* magnetic field signal.

**Figure 6 sensors-23-07519-f006:**
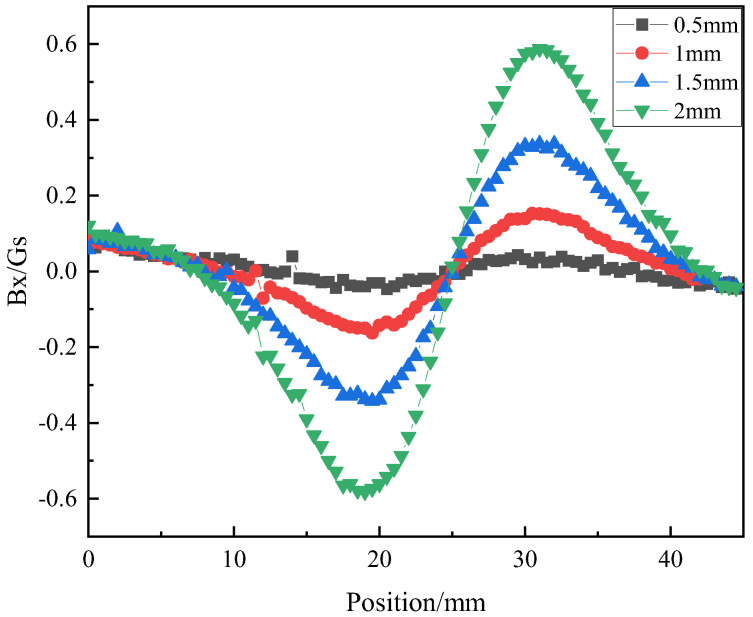
The relationship between girth weld forming height and Bx magnetic field signal.

**Figure 7 sensors-23-07519-f007:**
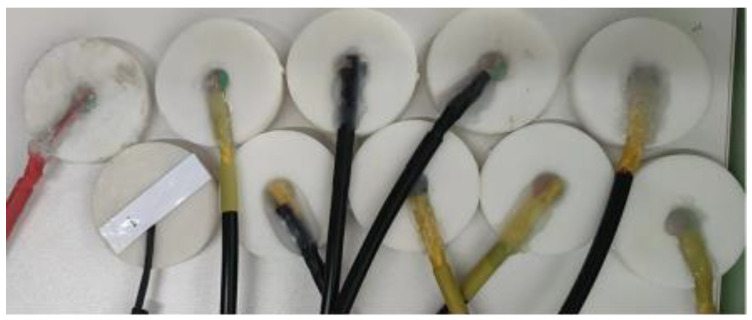
Probe structure.

**Figure 8 sensors-23-07519-f008:**
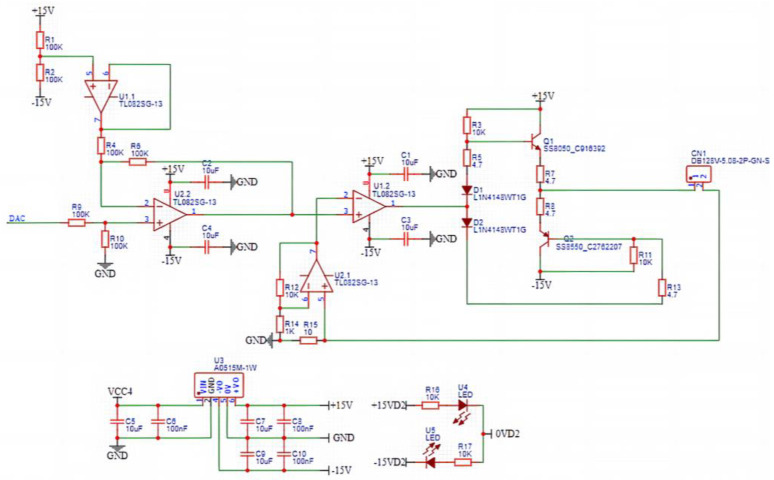
Current-type signal generation circuits.

**Figure 9 sensors-23-07519-f009:**
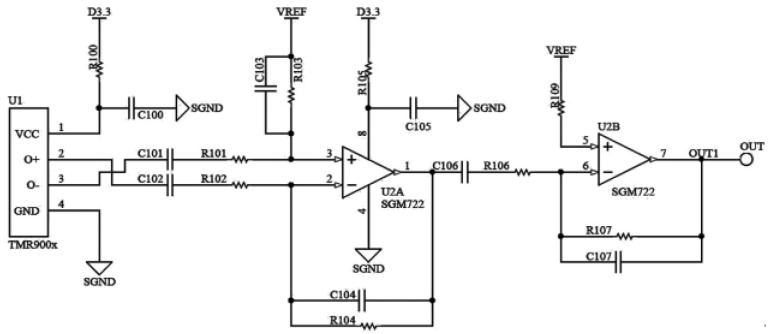
Amplification filter circuit.

**Figure 10 sensors-23-07519-f010:**
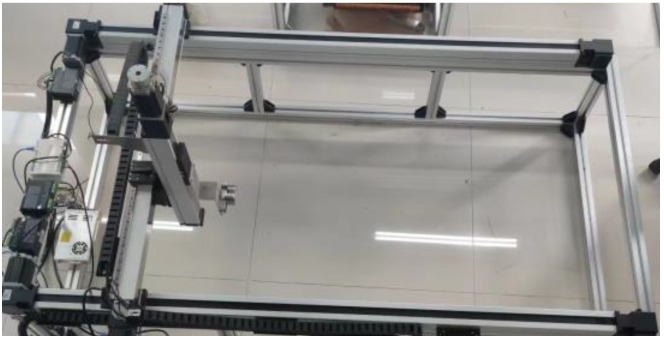
Three-axis linkage experimental platform physical map.

**Figure 11 sensors-23-07519-f011:**
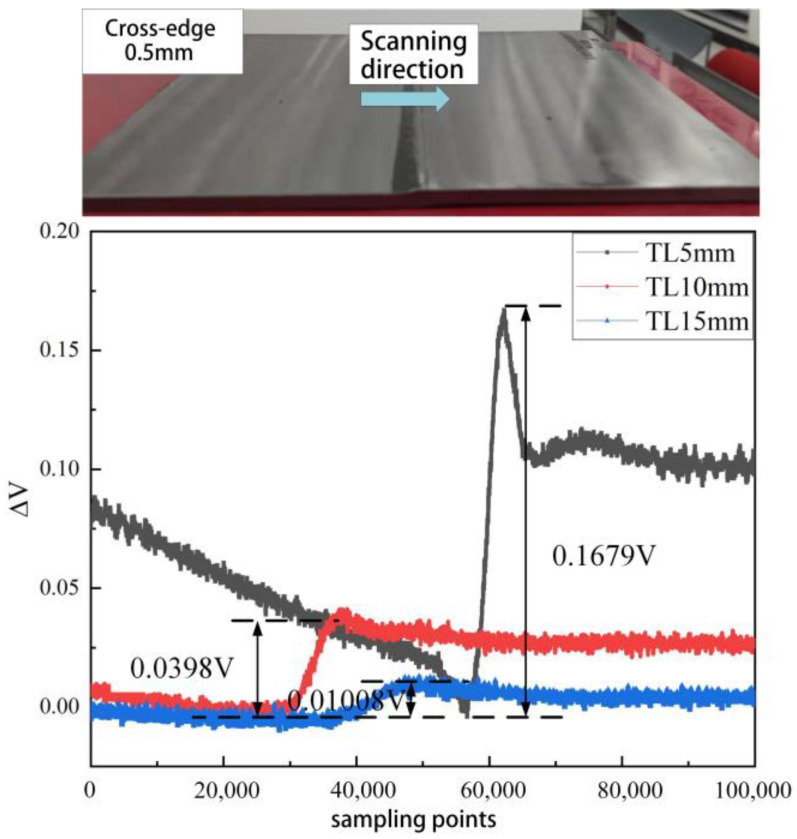
0.5 mm welding cross-edge response signal.

**Figure 12 sensors-23-07519-f012:**
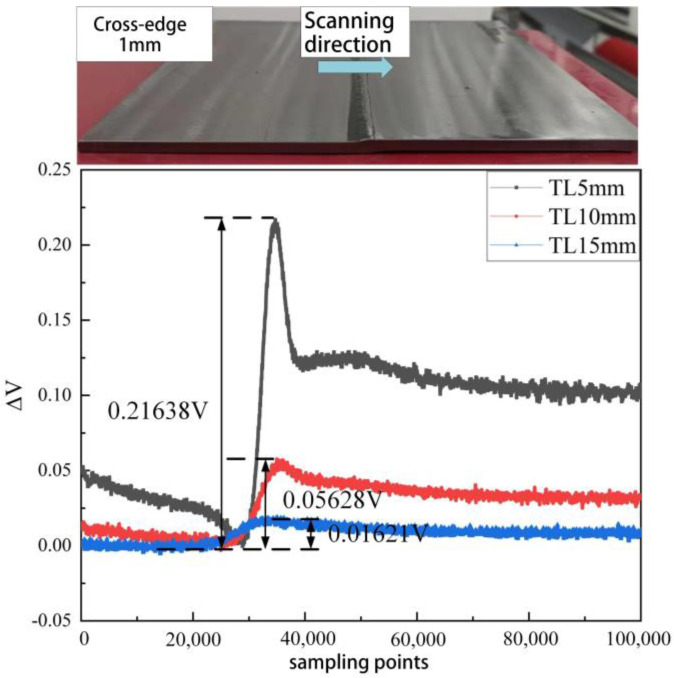
1.0 mm welding cross-edge response signal.

**Figure 13 sensors-23-07519-f013:**
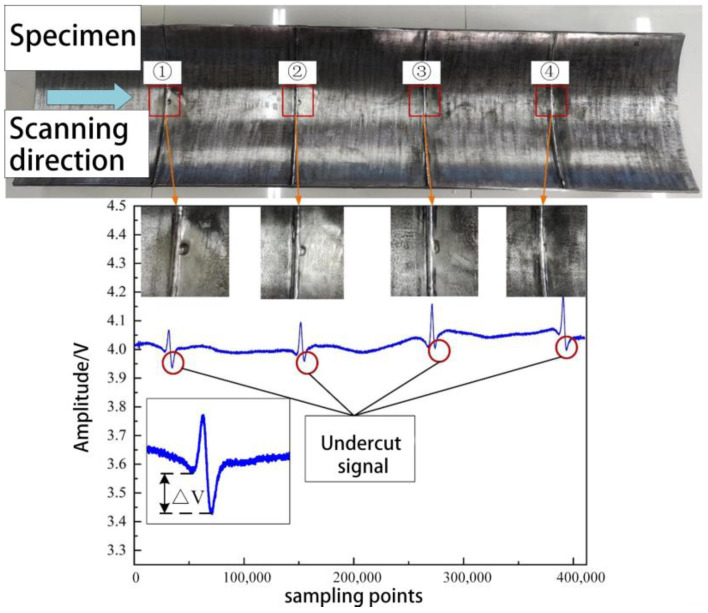
Abnormal weld formation—response signal of bite edge defect.

**Figure 14 sensors-23-07519-f014:**
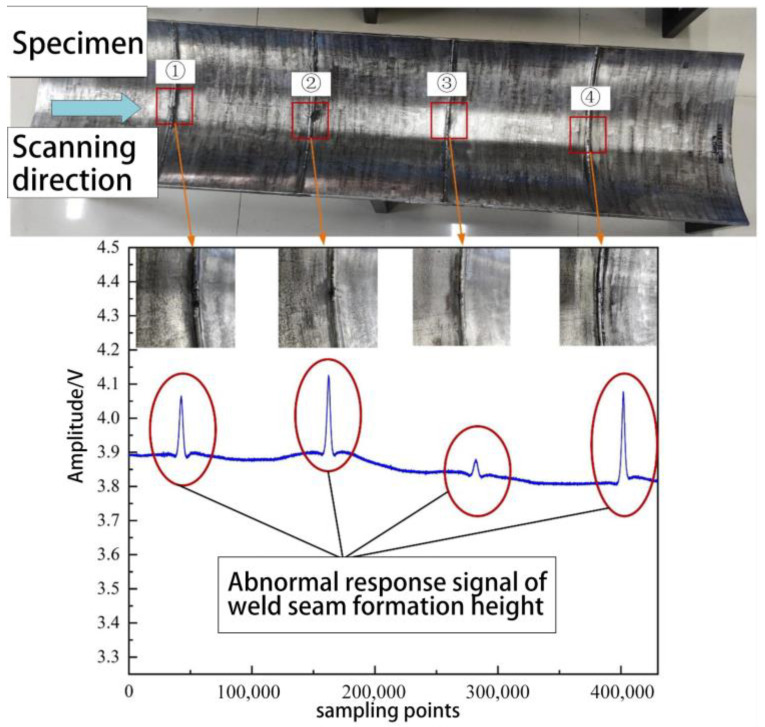
Abnormal response signal of girth weld formation height.

**Figure 15 sensors-23-07519-f015:**
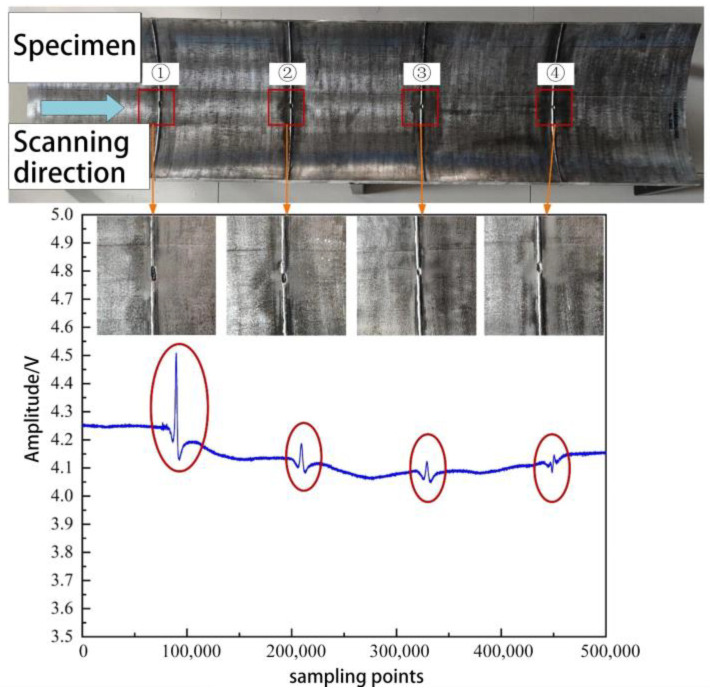
Abnormal girth weld formation—root concave defect response signal.

**Table 1 sensors-23-07519-t001:** Simulation model size parameters.

Name	Type	Value
specimen	Length	160 mm
	thickness	10 mm
coil	External diameter	20 mm
	Internal diameter	12 mm
	Turns	400
TL	15 mm	

**Table 2 sensors-23-07519-t002:** Material parameters of simulation model.

Region	Material	Electrical Conductivity (S·m^−1^)	Relative Permeability
Coil	Copper	5.995 × 10^7^	0.99
Specimen	45#steel	7.58 × 10^6^	1496
Air	Vaccum	1	1
Solution Region	Vaccum	1	1

**Table 3 sensors-23-07519-t003:** Solver parameters.

Caculation Step (max)	Error (%)	Iteration-Wise Encryption Fragmentation Unit Ratio (%)	Caculation Step (min)	Nonlinear Residue	Excitation Frequency (Hz)
10	0.5	50	4	0.0001	1000

**Table 4 sensors-23-07519-t004:** Relationship between Distortion Degree and Lift-off Height and Misalignment.

Misalignment	TL	Distortion ΔV
0.5 mm	5 mm	164.5 mV
	10 mm	51.9 mV
	15 mm	10.08 mV
1 mm	5 mm	216.38 mV
	10 mm	56.28 mV
	15 mm	16.21 mV

**Table 5 sensors-23-07519-t005:** Distortion Magnitude of Weld Edge Defect Signals.

Number	Length × Width × Deep	Distortion ΔV
①	8 mm × 10 mm × 2 mm	55 mV
②	8 mm × 10 mm × 1 mm	17 mV
③	6 mm × 10 mm × 1 mm	24 mV
④	4 mm × 10 mm × 1 mm	45.63 mV

## Data Availability

Data available on request due to restrictions e.g., privacy or ethical.

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
