# Peer review of "Research on Internal Shape Anomaly Inspection Technology for Pipeline Girth Welds Based on Alternating Excitation Detection"

_sensors, 2023, doi:10.3390/s23177519_

Round 1
Reviewer 1 Report
General Comments
The paper is poorly written, and difficult to understand. There seems to be very little connection between the calculations and the experiments. For instance, the experimental signals are never compared to the calculated signals. I would have liked to see more analysis to determine if the signal arises primarily from eddy currents or heterogeneities in the magnetic susceptibility. The experiments are useful in that there clearly is a signal that arises from defects in the sample. However, the article doesn’t do a good job explaining how and why the signal looks like it does. Also, it is not clear to me what the actual structure of “bite edge” and “weld seam” and other defects are. Some sort of better correlation of defect structure to experimental signal would make the article much stronger.
Specific Comments
Line 3: The title contains the word “girth” but it never appears in the text itself. What does it mean?
Line 36: Very strange to say “approximately” and then give a value to four significant figures. Also, I assume “104” means 10 raised to the 4th power, or 10000. I would just write “approximately two million kilometers”.
Line 36: “and it is…”. I assume “it” refers to total millage, not millage under construction. It is not clear.
Line 156: Equation 1 is confusing. Shouldn’t it be “del cross [(del cross A)/mu] = J” rather than “del cross [A/mu] = J”?
Line 161: Why isn’t there also a term on the right-hand-side of Eq 2 that includes “sigma grad V” where V is the voltage. In general, unless the applied electric field is somehow confined to have no component perpendicular to the pipeline wall, there will be a grad V term also.
Line 165: “at detection point A1”. Please show A1 and detection coil in Fig. 3
Line 172: “of the derived magnetic field”. What do you mean by “derived”?
Line 175: Can you give a reference for where you got Eq. 4? If not, then we need some sort of derivation or at least explanation of how you derived it.
Lines 180-182: After Eq. 4 you define many of the variables, but three that you define (H1, mu_r, and d) are not in Eq. 4, but other variables that are in Eq. 4 are not defined (z_1, z_2, z_0, r, r_0, r_1, r_2).
Line 190: I don’t understand what is the difference between a “defect” and a “discontinuity”
Line 202: You perform finite element calculations, so do you even use Eqs. 4 and 5? If you don’t use them, why show them?
Line 222: You say “since the curvature… is only five times greater…”. The “only” makes me think five times greater is rather small, and I am expecting curvature to play a big effect. Then, suddenly, you say “the impact of… curvature is minimal.” Why say “only” if what you really mean is that five times is big enough to ignore?
Line 222: By “curvature” do you mean “radius of curvature”? Curvature does not have dimensions of length.
Line 226: “axial symmetry of both the pipeline and the probe”. The pipeline and the probe don’t have axial symmetry along the same axis, so it seems your argument would support having a 3D simulation. Confusing.
Line 227: “Considering the high penetration capability of the internal detector”. I don’t understand why a high “penetration capability”, whatever that means, should lead to a 15 mm probe-to-pipeline separation. Explain better how you choose 15 mm. Wouldn’t you be better off with, say, 1.5 mm?
Line 229: What does the purple arrow and “V” in Fig. 4 mean?
Line 232: Table 1 contains “TL” but this variable has not been defined yet.
Line 239: Shouldn’t “106” and “107” be “10^6” and “10^7” with superscripts?
Line 239: The units of conductivity are given as “S m-1”. The authors need to learn how to use superscripts.
Line 244: What are “natural” boundary conditions?
Line 246: Table 3 lists several quantities that are never defined. I have no idea what they mean.
Line 248: The heading of Sec. 3.2 uses the term “molding” but that term is never used anywhere else in the article.
Line 249: We need some kind of better description of how a “seam” is implemented in the calculation. Is it represented as a change of conductivity, a change in magnetic susceptibility, or what? Is it a narrow discontinuity, like a crack, or something else? The main topic of the paper is analyzing welding defects, so a detailed description of the defect is essential.
Line 252: Does “Gs” mean “Gauss”? The usual abbreviation for Gauss is “G”.
Line 260: The authors say a “significant change” but is it? The magnetic field variation caused by the defect is less than 1%. Is that even measurable? I don’t know what is the noise in the system.
Line 287: “Figure 4.2”. But the figures are numbered 1, 2, 3, etc. There is no “4.2”.
Line 297: “The model of the magnetic sensor is TMR2503”. If you give a model number, you really must say who the manufacturer is.
Line 298: It is confusing to mix “Oe” and “T” units (and “G”). Choose one and stick with it.
Line 324: “Delta By” does not have units of mV. How do we convert to Tesla (or Gauss, or Oe, or whatever you are using for magnetic field).
Line 336: These values listed in Fig. 10 are given to four or five significant figures. The noise level is more than 1%. Are not those extra significant figures just garbage?
Line 342: “Table 5.1”. Do you just mean “Table 5”?
Line 352: What is a “forming height”?
The English grammar is not too bad. It is more the clarity of the writing, and the organization of the paper. Words are used but not defined. Different parts of the paper are not related to each other well.
Reviewer 2 Report
Dear authors,
I have read your valuable paper carefully. Pipeline systems ensure sustainable transportation of large quantities of oil and gas over long distances. Today, in terms of the green transition, we talk a lot about transporting carbon dioxide for CCS/CCUS projects or transporting large quantities of hydrogen. Regardless of whether we want to use existing pipelines for transportation or build new ones, pipeline inspection is equally important. Although the topic is very interesting and up-to-date, there are many small things that reduce the final quality of the work and need to be fixed. Below I send you a suggested set of corrections that should definitely be considered before publishing the manuscript:
General:
- the quality of all figures in the text of your manuscript should be improved (probably a slight enlargement of the figures will solve this problem),
- please insert spaces whenever necessary (e.g. between value and unit of measure, before quotations in the text of the manuscript, etc.),
- the entire text of the manuscript must be brought in line with the Instructions for Authors,
- please use the spelling in the third person,
- use lower or upper case letters,
- when we consider internal inspection of the existing pipeline, have you considered the impact of the medium within the pipeline? (see line 362)
Line 5: First capital letter in Mr. Chen's last name;
Line 35: 10 with exponent 4 instead of 104 km;
Line 41: the average “annual” cost?
Line 158: J0 or J with index 0?
Line 248: you have chapter 3.2. but you don’t have chapter 3.1 before;
Line 232: what does TL stand for? Sigma is the part of the unit of measurement or?
Line 251: What does By and Bx stand for?
Line 252: position of the...?
Line 287: Figure 7. instead of figure 4.2,
Line 298: Exponent -4,
Line 302- 305: The sentence is a bit unclear,
Line 307: diagram or figure?
Line 309-314: the experimental setup should be explained in more detail;
Line 326-327: figures 9 and 10 show a plate, and the explanation talks about a welded pipe;
Line 331: if the scanning direction is from left to right, the signal of the probe outputs increases;
Line 342: table 5.1. or only 5?
Lines 346-347: the sentence is a bit unclear;
Line 348: please add information about the diameter and wall thickness of the used pipe in the experimental part of your investigation;
Line 380: can you explain the difference between figure 13 and 11, since both figures have the same caption.
Yours sincerely
Reviewer
Round 2
Reviewer 1 Report
The article is improved from the original. Most of the specific comments I made in the original review were addressed.
The authors changed Eq. 2 as I suggested, but they did in incorrectly. The added term should be the gradient of V, not the time derivative of V. See any good E&M textbook. The authors changed the equation, but never mentioned in the text what V is. It would be useful to know if the author’s finite element method even takes V into account.
none
